# Association of NT-proBNP and sST2 with Left Ventricular Ejection Fraction and Oxidative Stress in Patients with Stable Dilated Cardiomyopathy

**DOI:** 10.3390/biomedicines12040707

**Published:** 2024-03-22

**Authors:** Elżbieta Lazar-Poloczek, Ewa Romuk, Wojciech Jacheć, Karolina Wróbel-Nowicka, Agata Świętek, Celina Wojciechowska

**Affiliations:** 12nd Department of Cardiology, Faculty of Medical Sciences in Zabrze, Medical University of Silesia, 10, M.C-Skłodowska St., 41-800 Zabrze, Poland; lazarelzbieta@gmail.com (E.L.-P.); wjachec@sum.edu.pl (W.J.); 2Department of Biochemistry, Faculty of Medical Sciences in Zabrze, Medical University of Silesia, 19, Jordan St., 41-808 Zabrze, Poland; eromuk@sum.edu.pl; 3Medical Laboratory in Specialistic Hospital in Zabrze, 10, M.C-Skłodowska St., 41-800 Zabrze, Poland; karolina.wrobel@diag.pl; 4Department of Medical and Molecular Biology, Faculty of Medical Sciences in Zabrze, Medical University of Silesia in Katowice, 19, Jordan St., 41-808 Zabrze, Poland; agata.swietek@sum.edu.pl; 5Silesia LabMed Research and Implementation Center, Medical University of Silesia in Katowice, 19, Jordan St., 41-808 Zabrze, Poland

**Keywords:** NT-proBNP, sST2, oxidative stress, heart failure, dilated cardiomyopathy

## Abstract

The aim of this study was to analyze the relationship between levels of sST2, NT-proBNP and oxidative stress markers in patients with reduced ejection fraction (HFrEF) due to non-ischemic cardiomyopathy. A total of 88 patients with HFrEF were divided into four groups based on left ventricular ejection fraction (≤25% and >25%) and NYHA functional class (group 1—LVEF > 25% and NYHA class I or II; group 2—LVEF > 25% and NYHA class III or IV; group III—LVEF ≤ 25% and NYHA class I or II; group IV—LVEF ≤ 25% and NYHA class III or IV). In 39 (44.32%) patients LVEF was reduced below 25%, and 22 of them (56.41%) were in NYHA functional class III/IV. Of the 49 (55.68%) patients with LVEF ≥ 25%, only 18.37% were in NYHA functional class III/IV (*p* < 0.001). Patients with LVEF ≥ 25% had lower levels of NT-proBNP, total oxidant status (TOS), total antioxidant capacity (TAC), and oxidative stress index (OSI). The levels of NT-proBNP but not sST-2 correlated positively with NYHA functional class (*p* < 0.001) and negatively with LVEF (*p* < 0.001). The levels of sST-2 were associated with increased TAC (*p* = 0.009) and uric acid (*p* = 0.040). These findings indicate that only NT-proBNP was related to the severity of heart failure, whereas sST2 correlated with total antioxidant capacity. Therefore, in stable patients with HFrEF due to dilated cardiomyopathy, sST2 may be an additional biomarker reflecting the redox status, but not the severity of heart failure.

## 1. Introduction

Despite a significant decline in mortality rates for cardiovascular disease, the onset of heart failure (HF) symptoms is predictive of a poorer survival outcome, and HF remains one of the most common causes of death globally [1,2]. Recent trials with sodium–glucose transport protein inhibitors (SGLT2i), also called flozins (e.g., dapagliflozin, empagliflozin, canagliflozin), have shown that one-year cardiovascular and all-cause mortality in heart failure with reduced EF (HFrEF) are high, and equal approx. 8 and 10% despite optimal pharmacological treatment [3,4]. The classification system most commonly used to assess limitations of physical activity is the New York Heart Association (NYHA) Functional Classification, which places patients in one of four categories [5]. Patients with marked limitations of physical activity or symptoms at rest (functional class III and IV) had a poorer prognosis than patients with no or slight symptoms during ordinary activity (class I or II), and did not show significant benefits derived from adding flozins to therapy. However, analysis in other subgroups with more advanced HF (e.g., more reduced ejection fraction and increased NT-proBNP levels) revealed that patients with LVEF < 30% and NT-proBNP above the median benefited more from therapy with empagliflozin [4]. Similarly, treatment with dapagliflozin was more effective in patients with LVEF below the median [3]; thus, the results were not consistent with those for the advanced NYHA class patients. Moreover, heart failure is characterized by a variety of etiologies and underlying pathophysiological processes that may have an influence on further survival in individual patients. One of the most common causes of HF and the most frequent cause of heart transplantation is dilated non-ischemic cardiomyopathy (NICM) [6]. Dilated cardiomyopathy is characterized primarily by enlargement of the left ventricle (LV) and impaired systolic function. Although there are newer techniques for the assessment of left ventricular dysfunction, left ventricular ejection fraction (LVEF) assessed via echocardiography remains the most important clinical variable. LVEF is a strong independent measure of LV systolic function, whereas NYHA classification based on patient-reported symptoms and functional capacity is subjective. Nevertheless, NYHA classification is still one of the simplest tools used by clinicians to rapidly assess functional status during exercise [7,8,9,10,11]. Moreover, this classification has been found useful for establishing the prognosis of patients with heart failure and guiding approval for therapy on its basis [12]. However, NYHA classification remains subjective and poorly reproducible [13]. Most large randomized trials in patients with HFrEF found both the LVEF and the NYHA class to be strong independent predictors of cardiovascular outcomes and all-cause mortality (e.g., MERIT, CHARM, PARADIGM-HF). Just like NYHA class, LVEF remains the eligibility criterion for pharmacotherapy and cardiac electrical therapy in HF [14,15,16]. 

NT-proBNP is an inactive protein, N-terminal prohormone of brain natriuretic peptide released from ventricular cardiomyocytes in response to increased wall tension [17]. NT-proBNP is useful in both the diagnosis and prognosis of heart failure, and NT-proBNP is also a strong predictor of death in acute and chronic heart failure as well as short- and long-term mortality in patients with suspected or confirmed unstable cardiovascular disease [18,19,20]. Natriuretic peptides (ANP and BNP) and high-sensitivity troponins (hsTn) are widely used as biomarkers in heart failure diagnosis [19,20,21], which is included in the guidelines of the European Society of Cardiology and American Heart Association (AHA) for the diagnosis and treatment of HF [1,22]. Despite many studies focusing on novel biomarkers [23,24,25,26,27] and their additive value in the diagnosis of HF, none of them have shown clinical utility.

One of the most thoroughly studied biomarkers linked to the inflammation and fibrosis processes is the suppression of tumorigenicity 2 (ST2). ST2 is one of the members of the interleukin-1 family and exists in four isoforms, two of which are clinically significant: transmembrane ST2L, and soluble sST2 circulating in the human serum (soluble ST2). Interleukin 33 (IL-33) has been identified as a functional ligand for ST2. The ST2 protein is involved in the immune response and is secreted in response to mechanical stretching. The mechanical stretching of cardiomyocytes and cardiac fibroblasts induces an increase in IL-33 release from cytoplasmic vesicles [28] and in the expression of ST2L and sST2 [29,30]. ST2L is a transmembrane isoform, which, by binding to IL-33, exerts anti-inflammatory and anti-fibrotic effects in the impaired heart through antagonizing the effects of angiotensin II and catecholamines. sST2 represents a soluble form of the protein that acts like a decoy receptor [30], prevents IL-33 from binding to the ST2L isoform and inhibits cardioprotective biochemical pathways initiated by IL-33. The biological effect of ST2 depends on the balance between transmembrane and soluble forms of the molecule. Evidence shows that sST2 plays an important role in the structural and functional remodeling of the left ventricle and its fibrosis, and may provide new insights into the diagnosis, prognosis and treatment of cardiovascular disease [31,32]. Soluble ST2 has recently emerged as a marker of mechanical strain [29,32,33,34]. Elevated concentrations of sST2 have been found to be associated with both acute and chronic HF [23,35,36,37,38]. sST-2 is strongly linked to remodeling, which is a key process in the development of dilated cardiomyopathy. Nonetheless, contrary to cardiac-specific markers like NT-proBNP and hsTn, levels of sST2 are not only reflected in cardiac dyspnea, but also by inflammatory disorders or cancer [31,39]. Thus, the protein lacks cardiac and HF specificity, being simultaneously a strong prognostic biomarker. Considering the above, the clinical value of sST2 in heart failure remains unclear.

Oxidative stress plays an important role in the pathophysiology of heart failure. Over the past decade, there has been interest in both oxidative stress and inflammation as potential contributors to heart failure progression. Oxidative stress is caused by an imbalance between the production and accumulation of reactive and nitrogen species (ROS/RNS) and the ability of the biological system to detoxify these reactive species. ROS are generated as a result of myocardial hypoxia [40]. Furthermore, this insufficient antioxidant protection may promote the dysfunction of cardiac mitochondria, which plays a pivotal role in heart failure progression [41,42].

The excessive precipitation of ROS from the mitochondria, with other mechanisms including xantine oxidase, lipoxygenase, reduced nicotinamide adenine dinucleotide phosphate (NADPH) oxidases, cytochrome P450 enzymes and peroxydases, may promote cardiomyocyte hypertrophy, encourage fibrosis and Ca^2+^ overload, and initiate apoptosis and the dysregulation of the inflammatory response [43,44]. All these phenomena are of great importance in the development of heart failure.

It is noteworthy that patients with NICM remain frequently asymptomatic at early stages because of the compensatory mechanisms; therefore, it is essential to identify prognostic biomarkers that are easy to detect. Because of the lack of a high correlation between NYHA and LVEF, we attempted to assess the usefulness of the established and novel biomarkers of the biomechanical strain (NT-proBNP and sST-2) in predicting LVEF in patients with stable dilated cardiomyopathy presenting with mild and severe HF symptoms (NYHA I/II and III/IV). Additionally, we decided to analyze the relationship between the severity of HF (NYHA class, LVEF) and biomarkers of biomechanical cardiovascular strain and the intensity of oxidative stress using a simple test to estimate oxidative stress index (OSI). We also checked the levels of non-enzymatic endogenous antioxidants such as bilirubin and uric acid, which are frequently measured in laboratory tests.

## 2. Patients and Methods

A total of 88 patients (17 female), mean age 50.3 ± 17.5 years, with heart failure with reduced ejection fraction, diagnosed as non-ischemic dilated cardiomyopathy (NIDCM) according to the WHO criteria, were included in the study [45]. Patients were admitted to the 2nd Department of Cardiology for follow-up as potential heart recipients. Those with clinical deterioration or changes in pharmacological therapy in the last 4 weeks were excluded from further study. Other exclusion criteria included uncontrolled arterial hypertension, significant renal or hepatic insufficiency, *chronic obstructive pulmonary disease*, chronic inflammatory or infectious disorders, and diagnosis of cancer in the last five years. All patients were treated with optimal-dose angiotensin-converting enzyme inhibitors or angiotensin receptor blockers, beta-blockers, mineralocorticoid receptor antagonists, digitalis and diuretics according to the European Cardiology Society guidelines before the introduction of ARNI and SGLT-2 [46]. The study protocol was approved by the Bioethics Committee of Medical University of Silesia (PCN/0022/KB1/23/II/20). A written informed consent form was obtained from all enrolled patients. All patients underwent physical examination. The functional capacity of each patient was assessed using the NYHA classification. Data regarding comorbidities, implanted electrical devices and pharmacological therapy were obtained. Weight and height were measured to calculate body mass index (BMI) and body surface area (BSA). Arterial pressure was measured according to the Korotkoff method. Electrocardiograms were analyzed to detect atrial fibrillation, left bundle brunch block (LBBB) and right bundle brunch block (RBBB). Echocardiography was performed as recommended by the American Society of Echocardiography Committee (GE, VIVID 7, Horten, Norway and 3S Sector Transducer). Echocardiograms were assessed by two experienced investigators independently (interobserver variability was <8%). The final decision was made through consensus. Left ventricular end-diastolic volume (EDV) and end-systolic volume (ESV) were obtained from the apical 4- and 2-chamber views by the modified Simpson’s method. Left ventricular ejection fraction (LVEF) was calculated in the standard manner to assess ventricular systolic function [47]. A sample size of 10 mL of blood was collected from the peripheral vein (usually antecubital) after overnight fasting at the time of medical examination. Serum was separated by centrifugation at 1500× *g* for 10 min at 4 °C, and then transferred into 1 mL cryotubes and stored at −70 °C for later analyses. sST2 levels were determined using enzyme-linked immunosorbent assay kits (ELISA) and the Quantikine Human sST2/IL-33 R Immunoassay (R&D Systems, Minneapolis, MN, USA), with assay range 31.3–2000 pg/mL and sensitivity 13.5 pg/mL. The intra-assay and inter-assay precisions were about 4.8 and 6.3 on average, respectively. Calibration of the assay was performed according to the manufacturer’s recommendations and values were normalized to a standard curve. NT-proBNP was measured by an electrochemiluminescence immunoassay (Cobas 6000e501, Roche Diagnostics, Mannheim, Germany). Blood hemoglobin, sodium, uric acid, transaminases, bilirubin, creatinine concentrations and serum lipid profile were measured using routine techniques (Cobas Integra 800 Roche Diagnostics, Mannheim, Germany). The total oxidant status (TOS) was measured by the spectrophotometric method developed by Erel [48]. In Erel’s method, the oxidizing materials contained in the sample lead to the oxidation of Fe^2+^ ions to form Fe^3+^. The reaction proceeds in an acidic environment and involves measuring the color intensity of Fe^3+^ ion complexes with xylenol orange. TOS was expressed in µmol/L. Total antioxidant capacity (TAC) was also measured by Erel’s spectrophotometric method [49]. This method is based on the 2,2-azinobis (3-ethylbenzo-thiazoline-6-sulfonate) (ABTS+) reaction. A colorless molecule, reduced ABTS, is oxidized to a blue-green ABTS+. After mixing the colored ABTS+ with any substance that can be oxidized, it is reduced to its original, colorless ABTS form again and the reacted substance is oxidized. TAC was expressed in mmol/L. Oxidative stress index (OSI) was calculated as the percentage ratio of TOS to TAC. OSI (%) = [TOS (µmol/L)/TAC (µmol/L)] × 100 [50].

## 3. Statistical Analysis

To assess the normality of the continuous data, the Shapiro–Wilk test was used. Categorical data were presented as numbers and percentages. Because most of the continuous data were non-normally distributed, they were expressed as median and lower (Q1) and upper (Q3) quartiles. The entire group of patients were divided into four subgroups based on left ventricular ejection fraction (≤25% and >25%) and NYHA functional class (group 1—LVEF > 25% and NYHA class I or II; group 2—LVEF > 25% and NYHA class III or IV; group III—LVEF ≤ 25% and NYHA class I or II; group IV—LVEF ≤ 25% and NYHA class III or IV). The Kruskal–Wallis ANOVA test was used for the comparison of the sST2 and NT-proBNP concentrations and the redox parameters presented in Figure 1, Figure 2 and Figure 3. Non-parametric tests (Chi^2^ test with Yates correction or the unpaired Mann–Whitney U test, as appropriate) were used to compare baseline parameters between the groups divided based on LVEF (≤25% and >25%). To determine the correlations between sST2 and NT-proBNP levels, as well as clinical, echocardiographic and laboratory data, the Spearman r test was used. Predictors of LVEF > 25% were determined using univariate and multivariable linear regression analysis. The variables achieving *p* < 0.1 in the Mann–Whitney U test and Chi2 test were included in the univariate regression analysis. Furthermore, variables with *p* < 0.1 in the univariate analysis were selected for the multivariable regression models. The results were considered statistically significant if *p* < 0.05. Statistical analysis was performed using the Statistica 13.3 (TIBCO, Kraków, Poland).

## 4. Results

### Clinical and Laboratory Characteristics

Of all patients admitted for routine assessment according to our Heart Transplant Qualification Protocol based on the Cardiac Transplantation Guidelines [51], 88 subjects with heart failure of non-ischemic origin (no significant atherosclerotic changes on coronary angiography) diagnosed as dilated cardiomyopathy (DCM), with a mean age of 50.36 (41.2–56.6) years (17 females), were finally enrolled in the study. All patients were clinically stable without changes in the treatment for at least one month. In 39 (44.32%) patients, LVEF was reduced below 25%; in this subgroup, 22 (56.41%) were in the NYHA functional class III/IV. Of the 49 (55.68%) patients with LVEF ≥ 25%, only 18.37% were in NYHA functional class III/IV (*p* < 0.001). The majority of patients were treated with β-blockers (96.6%) and either an angiotensin-converting enzyme (ACE) inhibitor (97.7%) or an angiotensin receptor blocker (36.4%) and a mineralocorticoid receptor antagonist (90.9%). Some of the patients received digitalis (63.6%), loop and/or thiazide diuretics (65.9% and 17.0%, respectively). The incidence of comorbidities and ECG abnormalities was similar in both groups, whereas in the LVEF < 25% group, both ventricles and atria were larger. The patients with LVEF ≥ 25% had significantly lower levels of NT-proBNP, TAC, creatinine, uric acid and bilirubin, and higher levels of sodium, TOS and OSI (Table 1).

Levels of NT-proBNP and sST-2 in groups with LVEF < 25% and ≥25% based on NYHA classification are displayed in Figure 1; TAC in Figure 2, TOS in Figure 3 and OSI is shown in Figure 4.

Levels of NT-proBNP, but not sST-2, correlated positively with NYHA functional class, and left and right ventricular enlargement, and negatively with LVEF. Levels of NT-proBNP correlated negatively with hemoglobin and positively with bilirubin and uric acid concentrations. There was no correlation between NT-proBNP and sST-2. Levels of sST-2 correlated positively with TAC and uric acid concentrations. NT-proBNP did not correlate with oxidative stress markers but correlated positively with uric acid and bilirubin (Table 2).

Considering LVEF, there was a positive correlation between sST2 and NT-proBNP both in groups among patients with EF < 25% (r = 0.435, *p* = 0.006) and in the group of patients with EF > 25% (r = −0.146, *p* = 0.316). sST2, but not NT-proBNP, correlated positively with IL-6 (r = 0.460, *p* = 0.008) and TAC (r = 0.367, *p* = 0.021) among patients with EF < 25%. 

After dividing patients according to NYHA groups, we did not demonstrate any correlations between sST2 and NT-proBNP for groups NYHA I-II (*p* = 0.886) and NYHA III-IV (*p* = 0.224). sST2, but not NT-proBNP, correlated positively with IL-6 (r = 0.574, *p* = 0.002) and TAC (r = 0.534, *p* = 0.002) only in group NYHA III-IV. Nonetheless, there was no correlation between sST2, NT-proBNP and TOS or OSI in the individual groups. 

Univariate analysis indicated that higher NYHA functional class and lower levels of NT-proBNP, creatinine and uric acid, as well as higher levels of sodium, TOS and OSI, were the predictors of EF > 25%. Multivariable analysis showed that only NT-proBNP and OSI remained a significant predictor of EF > 25% (0.926, CI 0.868–0.989) (Table 3). 

## 5. Discussion

To our knowledge, this is the first study assessing the relationship between exercise tolerance, left ventricular ejection fraction and markers of mechanical and oxidative stress in patients with chronic heart failure with reduced ejection fraction due to nonischemic dilated cardiomyopathy. 

In this study, LVEF correlated with NT-proBNP but not with sST2, and lower NT-proBNP was an independent predictor of LVEF > 25%. NT-proBNP correlated with the volume of the left ventricle and left and right atrial area, but sST-2 was linked only to left atrial area, indicating that NT-proBNP is a stronger marker of cardiac wall stress than sST-2 in clinically stable patients. Similarly, Najarr et al. [52] found no correlation between sST-2 and LVEF, and a correlation between sST-2 and left atrial volume index, but only in HFpEF patients. Soluble ST2 appears to be a biomarker with a putatively beneficial role in acute HF, rather than in chronic HF. The Investigation of Dyspnea in the Emergency Department (PRIDE) study [53] analyzed patients admitted to the emergency department for acute dyspnea, and revealed that the concentration of sST-2 was significantly higher in patients with acute heart failure than in patients with non-cardiac dyspnea. However, the comparison of the diagnostic value of NT-proBNP and sST-2 showed that NT-pro BNP was a much better parameter in the diagnosis of acute HF. In another study [54], the investigators examined the relationship between sST-2 and clinical patients’ characteristics and prognoses. It turned out that baseline levels of sST-2 correlated with NYHA functional class, left ventricular ejection fraction and NT-pro BNP. Other clinical studies have shown that serum sST2 is increased after acute myocardial infarction and inversely correlates with ejection fraction [29]. Rehman et al. [34] found a weak correlation between sST-2 and ejection fraction among patients with acute HF. Notably, the investigators demonstrated that sST-2 predicted 1-year mortality in a similar way to NT-proBNP. However, it is worth noting that the mean LVEF in the study was 45%. Dimitropoulos et al. [55] found that sST2 levels were associated with HF status and the functional capacity of patients stratified according to NYHA classification. Indeed, patients with functional impairment were more likely to have higher sST2, although the levels of circulating sST2 were not associated with LVEF in patients with ischemic heart failure. In the present study, sST2 did not correlate with NYHA functional class. Interestingly, the results of studies measuring sST2 in patients with chronic heart failure are not consistent. Not all studies confirm an increase in sST2 levels that correlates with the severity of heart failure symptoms or LVEF. In the ValHEFT study [56], the investigators measured sST-2 in patients with systolic left ventricular dysfunction at baseline, at 4 months and at one year. As it turned out, the predictive value of baseline sST-2 levels was low and completely offset by NT-proBNP levels. In a study by Miftode et al. [57], there was no correlation between sST-2 levels and LVEF, despite a significant correlation with NT-proBNP. In their study, patients with stable chronic HF served as controls and had significantly lower sST2 levels than patients with acute HF (29.2 vs. 107.2 ng/mL).

In our study, the median sST-2 was even lower. Therefore, sST2 appears to be a more useful diagnostic biomarker in acute HF than in chronic stable HF. Most clinical studies have also shown the prognostic value of sST2 in chronic HF [58]. The sensitivity and specificity of sST2 as a predictor of hospitalization due to HF and cardiovascular death were similar to those of NT-proBNP and hsTn [59]. In our study we did not demonstrate statistically significant differences in the concentration of sST2 (*p* = 0.272) and NT-proBNP (*p* = 0.439) in the subgroups of patients with AF and/or LBBB. This may be due to the fact that 21 of 29 patients with LBBB received cardiac resynchronization therapy.

However, there are also some parameters that may affect sST2 or NT-pro BNP levels. Our findings are consistent with the observation that sST2, contrary to NT-proBNP, is less influenced by body mass index (BMI) [34,59,60] and estimated glomerulal filtration rate (eGFR). Thereby, theoretically, sST2 may be characterized by better specificity than NT-proBNP. However, Dieplinger et al. [61] and Mueller et al. [62] have shown that sST2 is strongly influenced by severe infection including pneumonia, sepsis or chronic obstructive pulmonary disease. Furthermore, sST2 is associated with gastric and breast cancer [35,63,64,65,66], and increased sST2 has been reported in patients with COVID-19 compared with non-COVID patients with chronic stable HF [57], which, contrary to expectations, means that sST2 cannot be considered as a cardiac-specific marker. The strong relationship between sST2 and NT-proBNP in both HFpEF and HFrEF was demonstrated by Najjar et al., but in their study only 5% of HFrEF patients were in NYHA classes I and II (65% in our study) [52]. 

In this study, we examined an association between oxidative stress, antioxidant parameters and LVEF in patients with dilated cardiomyopathy with varying degrees of exercise intolerance. TAC reflects the cumulative action of all antioxidants present in blood plasma [67]. TOS is usually used to determine the overall oxidation state of the body [48]. OSI is the ratio of TOS to TAC [68], and may be a more accurate, comprehensive measure of TAC and TOS. We demonstrated higher TOS in patients with LVEF ≥ 25%, which resulted in higher OSI in these patients. Although the highest TAC value was found in patients with LVEF < 25% and low functional class (NYHA III and IV), OSI was comparable to that in patients with EF < 25% and NYHA class I and II, and lower than in patients with LVEF ≥ 25%. OSI was a predictor of LVEF ≥25%, but only in univariate regression. Diraman et al. [69] demonstrated lower serum levels of intracellular antioxidants SOD, CAT, GSHP and glutathione reductase with ejection fractions lower than 25% among patients with dilated cardiomyopathy. On the other hand, Kamel et al. observed a generally statistically positive correlation between TAC and LVEF in patients with HFrEF [70]. In contrast, Mongirdienė et al. [71] revealed no significant differences between levels of oxidative (nitrotyrosine, dityrosine, protein carbonyl, malondialdehyde, oxidized HDL) and antioxidant (TAC, catalase) stress biomarkers and LVEF. However, it should be noted that patients in their study were stratified into two groups according to left ventricular ejection fraction (LVEF) values: HFrEF (<40%) and HFpEF (≥40%). We observed positive correlations for both ST-2 and NT-proBNP, and non-enzymatic antioxidants bilirubin and uric acid, the components of TAC (paradoxically, uric acid is an antioxidant, but it is produced in a reaction whose parallel products are free radical derivatives of oxygen). Surprisingly, we did not demonstrate a correlation between NT-proBNP and TAC in plasma. In our study, we demonstrated positive correlations between sST2 and TAC in both groups EF < 25% (r = 0.367, *p* = 0.021) and NYHA III-IV (r = 0.534, *p* = 0.002), which indicates the dependence of oxidative and inflammatory status on the severity of the disease. Therefore, the positive correlation between sST2 and TAC may reflect the former’s additional impact on antioxidant defense in HFrEF patients. Indeed, soluble ST2 (sST2) is an attractive molecule associated with inflammation, fibrosis and oxidative stress; however, the results available from clinical studies are inconsistent. We need to remember that sST2 is part of the IL-33/ST2L/sST2 signaling network. In the common model, sST2 is a decoy receptor for IL-33, thereby preventing the cellular actions of IL-33. However, there are several experimental reports that sST2 may have its own anti-inflammatory effects, e.g., direct effects on macrophages via the downregulation of Toll-like receptors, and the inhibition of LPS-induced IL-6 production in a human monocytic leukemia cell line [72,73]. 

Roy et al. [74] reported that, among patients with metabolic syndrome, increased levels of circulating sST2 correlated positively with inflammatory markers: IL-6, osteopontin, ICAM-1, myeloperoxidase and a classical reactive oxygen species marker nitrotyrosine (NT-Tyr). Furthermore, the investigators demonstrated a correlation between oxidative stress markers like 8-hydroxy-2′-deoxyguanosine (8-OHdG) and levels of hydrogen peroxide. On the other hand, there was no correlation between serum sST2 levels and TBARs or total nitric oxide. The sST2-related inflammatory cell recruitment may trigger oxidative stress, as demonstrated by the positive association with hydrogen peroxide. In the study by Zhang in HF patients, sST2 levels positively correlated with serum malondialdehyde, while they negatively correlated with the antioxidant superoxide dismutase activity, suggesting a role for elevated sST2 in enhanced redox status [74,75]. Matilla et al. [76], using proteomics and immunodetection approaches, demonstrated that sST2 downregulated mitofusin-1 (MFN-1), a protein involved in mitochondrial fusion, in human cardiac fibroblasts. In parallel, sST2 increased nitrotyrosine, protein oxidation and peroxide production. Also, the in vitro experiments show that sST2 increases oxidative stress and inflammation in human cardiac fibroblasts, leading to cardiac damage. This study reveals new information suggesting that sST2 may promote oxidative stress contributing to cardiac damage. Cohen et al. provided important information about the rapid termination of IL-33/ST2L activity via the oxidation of cysteine and disulfide bond formation in IL-33 in asthmatics [77]. The disruption of this inactivation in vivo leads to an enhancement of inflammation.

In conclusion, our findings indicate that in stable patients with HF due to dilated cardiomyopathy, NT-proBNP, but not sST2, was a useful predictor of LVEF based on biomechanical strain markers. Similarly, there was a high correlation between NT-proBNP and NYHA functional class. When analyzing oxidative parameters, both NT-proBNP and sST2 correlated positively with the non-enzymatic antioxidants such as uric acid and bilirubin. However, only sST2 correlated positively with TAC. Further research on a larger group of subjects is necessary to thoroughly explain the existing correlation between sST2 and oxidative parameters.

The most important limitation of the study is the small number of participants and the lack of a control group. The new studies will include patients already treated with ARNI and sGLT-2 inhibitors.

## Figures and Tables

**Figure 1 biomedicines-12-00707-f001:**
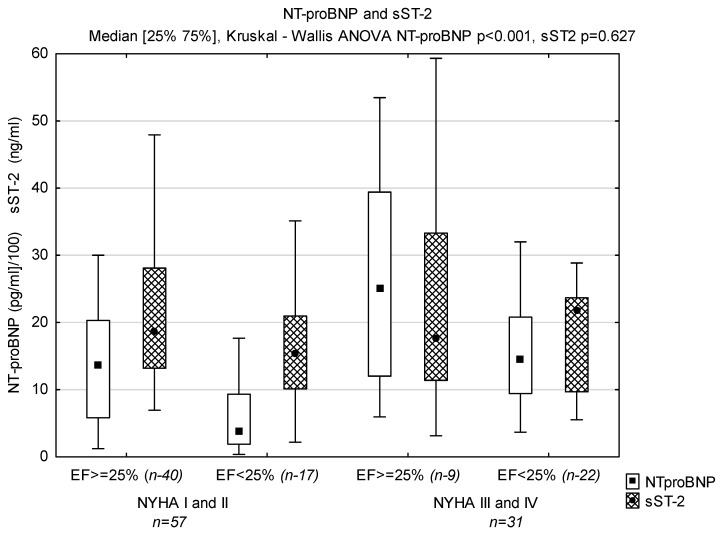
Levels of NT-proBNP and sST2 in groups divided according to NYHA functional class and ejection fraction (*n*—number of patients).

**Figure 2 biomedicines-12-00707-f002:**
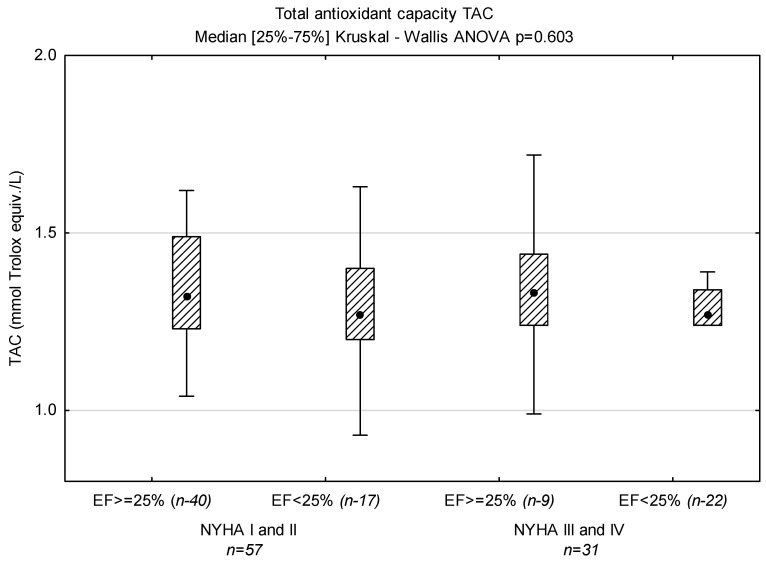
Levels of TAC in groups divided according to NYHA functional class and ejection fraction (*n*—number of patients).

**Figure 3 biomedicines-12-00707-f003:**
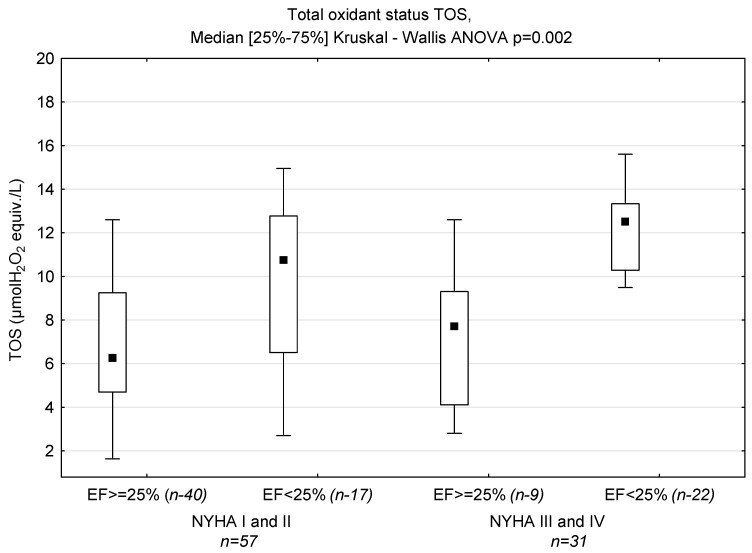
Levels of TOS in groups divided according to NYHA functional class and ejection fraction (*n*—number of patients).

**Figure 4 biomedicines-12-00707-f004:**
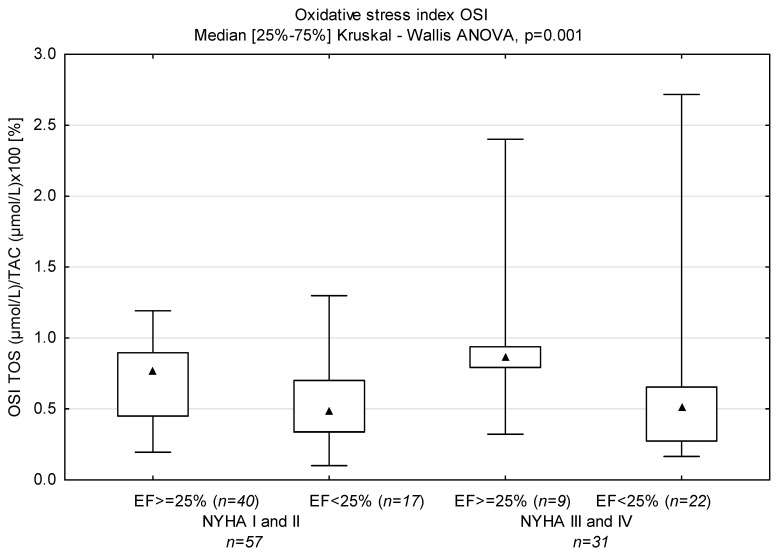
Levels of OSI in groups divided according to NYHA functional class and ejection fraction (*n*—number of patients).

**Table 1 biomedicines-12-00707-t001:** Comparison of clinical, echocardiographic and laboratory data.

	LVEF ≥ 25% *n* = 49	LVEF < 25% *n* = 39	*p*
	Median (Q1–Q3) *n*, %	Median (Q1–Q3) *n*, %	
Age, years	51.40 (43.15–56.70)	49.9 (37.90–56.00)	0.582
Female *n*, %	10 (20.40)	7 (17.94)	0.999
BMI, kg/m^2^	27.89 (24.33–30.69)	27.08 (23.31–29.91)	0.342
NYHA III-IV	9 (18.37)	22 (46.41)	<0.001
Arterial pressure (systolic), mm Hg	126.5 (112.0–142.8)	106.8 (93.00–126.0)	0.005
Arterial pressure (diastolic), mm Hg	78.50 (69.50–88.50)	71.25 (64.50–80.00)	0.046
Arterial pressure (mean), mm Hg	94.67 (84.75–105.6)	84.92 (71.67–93.33)	0.005
Heart rate/min	66.5 (62.4–81.2)	77 (64–88.8)	<0.008
Co-morbidity and ECG abnormalities			
AH, *n* [%]	13 [26.53]	9 [23.08]	0.764
Diabetes, *n* [%]	7 [14.29]	6 [15.38]	0.888
AF, *n* [%]	8 [16.32]	14 [35.89]	0.063
LBBB, *n* [%]	14 [28.57]	15 [38.46]	0.450
RBBB, *n* [%]	1 [2.04]	2 [5.12]	0.428
Echocardiographic data			
LVEF, %	30 (25–34)	19 (15–20)	<0.001
LVEDV, mL	177 (135–219)	220 (170–260)	0.009
LVEDD, mm	65 (59–69)	71 (65–74)	0.004
LA area, cm^2^	23 (19–27)	31 (25–37)	<0.001
RA area, cm^2^	17 (14–20)	23 (19–30)	<0.001
Laboratory data			
NT-proBNP, pg/mL	626.0 (258.0–1158)	1728 (786.0–3230)	<0.001
sST2, ng/mL	16.90 (10.00–21.78)	18.6 (11.39–29.78)	0.297
CRP, mg/L	5.91 (3.47–8.69)	5.51 (2.92–8.56)	0.916
Interleukin-6 pg/mL	3.29 (1.99–7.92)	2.61 (1.48–6.23)	0.478
Creatinine, mmol/L	76.90 (69.90–88.00)	86.6 (72.40–100.0)	0.032
sodium, mmol/L	137.0 (135.0–140.0)	136.0 (133.0–138.0)	0.040
hemoglobin, g/dL	14.40 (13.50–15.10)	13.60 (13.00–15.40)	0.420
uric acid, μmol/L	407.0 (317.0–472.0)	437.0 (385.0–522.0)	0.014
AspAT, IU/L	22.00 (17.00–33.00)	27.00 (20.00–34.00)	0.181
AlAT, IU/L	27.00 (20.00–47.00)	28.50 (19.00–49.00)	0.840
Bilirubin, µmol/L	12.50 (9.55–16.55)	16.50 (12.20–24.20)	<0.001
TAC mmol/L	1.39 (1.31–1.50)	1.48 (1.38–1.62)	0.029
TOS µmol/L	11.23 (6.67–12.88)	6.80 (4.50–9.31)	<0.001
OSI: TOS (µmol/L)/TAC (µmol/L) × 100 [%]	0.81 (0.58–0.92)	0.46 (0.29–0.68)	<0.001

BMI, body mass index; NYHA, New York Heart Association functional class; AH, arterial hypertension; AF, atrial fibrillation; LBBB, left bundle branch block; RBBB, right bundle branch block; LVEF, left ventricular ejection fraction; LVEDV, ventricular end-diastolic volume; LVEDD, left ventricular end-diastolic diameter; LA area, left atrial area; RA area, right atrial area; NT-proBNP, *N*-terminal pro-B-type natriuretic peptide; sST2, soluble suppression of tumorigenicity 2 *protein*; *CRP*, *C-reactive protein*; AspAT, aspartate aminotransferase; AlAT, alanine aminotransferase; TOS, total oxidant status; TAC, total antioxidant capacity; OSI, oxidative stress index.

**Table 2 biomedicines-12-00707-t002:** Correlations between clinical, echocardiographic, and laboratory data, as well as levels of sST2 and NT-proBNP.

	sST2		NT-proBNP	
	Spearman r	*p*	Spearman r	*p*
Age, years	0.071	0.509	−0.264	0.060
BMI, kg/m^2^	−0.056	0.607	−0.378	<0.001
NYHA, I-IV	0.119	0.267	0.582	<0.001
Arterial pressure (systolic), mm Hg	−0.072	0.552	−0.496	<0.001
Arterial pressure (diastolic), mm Hg	0.165	0.171	−0.273	0.022
Arterial pressure (mean), mm Hg	0.072	0.553	−0.377	<0.001
LVEF, %	−0.080	0.450	−0.501	<0.001
LVEDV, mL	−0.047	0.666	0.280	0.008
LVEDD, mm	−0.114	0.175	0.288	0.006
LA area, cm^2^	0.229	0.03	0.410	<0.001
RA area, cm^2^	0.039	0.741	0.511	<0.001
NT-proBNP, pg/mL	0.139	0.195		
sST2, ng/mL			0.139	0.195
CRP, mg/L	0.047	0.704	−0.200	0.070
Interleukin-6 pg/mL	0.081	0.466	−0.019	0.858
eGFR, mL/min	−0.159	0.138	−0.263	0.013
sodium, mmol/L	0.011	0.918	−0.204	0.056
hemoglobin, g/dL	0.101	0.346	−0.226	0.033
AspAT, IU/L	−0.023	0.736	0.074	0.488
AlAT, IU/L	−0.034	0.753	−0.063	0.531
uric acid, μmol/L	0.212	0.040	0.333	0.002
bilirubin, µmol/L	0.210	0.049	0.364	<0.001
TOS, µmol/L	−0.138	0.196	−0.122	0.256
TAC, mmol/L	0.276	0.009	0.018	0.869
OSI	−0.204	0.056	−0.095	0.359

BMI, body mass index; NYHA, New York Heart Association functional class; LVEF, left ventricular ejection fraction; LVEDV, ventricular end-diastolic volume; LVEDD, left ventricular end-diastolic diameter; LA area, left atrial area; RA area, right atrial area; NT-proBNP, *N*-terminal pro-B-type natriuretic peptide; sST2, soluble suppression of tumorigenicity 2 *protein*; *CRP*, *C-reactive protein*; *eGFR*, estimated glomerular filtration rate; AspAT, aspartate aminotransferase; AlAT, alanine aminotransferase; TOS, total oxidant status; TAC, total antioxidant capacity; OSI, oxidative stress index.

**Table 3 biomedicines-12-00707-t003:** Predictors of higher LVEF, results of univariate and multivariable linear regression.

Variable	Prediction LVEF ≥ 25%					
	Univariate Regression			Multivariable Regression		
	OR	95%	*p*	OR	95%CI	*p*
NYHA III-IV	0.274	0.125–0.600	<0.001	0.642	0.211–1.951	0.428
NT-proBNP, pg/mL	0.903	0.857–0.952	<0.001	0.926	0.868–0.989	0.020
creatinine mmol/L	0.971	0.948–0.996	0.019	0.984	0.953–1.016	0.303
sodium, mmol/L	3.940	1.196–12.98	0.022	2.401	0.524–11.01	0.252
uric acid, μmol/L	0.943	0.903–0.985	0.008	0.981	0.929–1.037	0.499
Bilirubin, µmol/L	0.974	0.939–1.012	0.174			
TAC mmol/L	0.316	0.038–2.672	0.283			
TOS µmol/L	1.200	1.059–1.361	0.004			
OSI	5.910	1.250–27.853	0.015	3.810	1.053–13.780	0.038

NYHA, New York Heart Association functional class; NT-proBNP, *N*-terminal pro-B-type natriuretic peptide; TAC, total antioxidant capacity; TOS, total oxidant status; OSI, oxidative stress index.

## Data Availability

The original data are available upon contact with the corresponding author.

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
