# Peer review of "Association of NT-proBNP and sST2 with Left Ventricular Ejection Fraction and Oxidative Stress in Patients with Stable Dilated Cardiomyopathy"

_biomedicines, 2024, doi:10.3390/biomedicines12040707_

Round 1

Reviewer 1 Report

Comments and Suggestions for Authors

The Authors performed an interesting analysis about the relationship between NT-proBNP and ST2 with left ventricular ejection fraction and oxidative stress in patients with stable dilated cardiomyopathy. The analysis is well conducted, the results are reasonable and appropriately discussed. I have no specific comments.

Author Response

Dear Reviewer

Thank you for reading the work carefully and for your positive assessment of the work.

We hope that the results of our research will be valuable and helpful in understanding the problems related to stable dilated cardiomyopathy.

Best regards

Ewa Romuk

Reviewer 2 Report

Comments and Suggestions for Authors

The aim of this study was to analyse the relationship between the level of sST2, NT-proBNP and oxidative stress markers in patients with reduced ejection fraction (HFrEF) due to non-ischemic cardiomyopathy. A group of 88 patients with HFrEF were divided into four groups depending on left ventricle ejection fraction (≤ 25% and >25%) and NYHA class (I-II or III-IV) values. sST2 was not different between the groups while NT proBNP increased with the severity. sST2 had poor correlations with LA area and TAC that did not differ between the groups.

They conclude with “in stable patients with HF due to dilated cardiomyopathy, NT-proBNP, but not sST-2, was a useful predictor of LVEF based on biomechanical strain markers. Similarly, a high correlation was found between NT-proBNP and NYHA class. However, only sST-2 concentration correlated with oxidative stress, and the positive correlation with TAC is surprising. Further research and determination of larger amounts of enzymatic and non-enzymatic antioxidants as well as oxidation products seem necessary.”

Many recent papers have studied sST2 and NT-proBNP in HFrEF. While NTproBNP is well established, sST2 is more recent and certainly looks for its relevance in the diagnostic area of HF in general. The use of a broad range population in the present did not help to find its possible position.

Abstract : Define TAC TOS and OSI

Introduction

Give a definition of NYHA classification and the justification of clusterization of I-II and II-IV

Define flozin and empagliflozin etc..

LVEF is a poor echocardiographic index. The study would have been more interesting with the GLS. Do you have the GLS ?

Did you measure systemic arterial pressure ?

Methods

Give the reference number of ethical committee and clinical trial

Define LBBB and RBBB

What is the echocardiographic machine? How many different sonographers?

Some medications, eg some beta-blockers, are known to have antioxidant properties? How did you take into account this point?

A control group without HF would have been of interest.

I supposed you measured sST2. Label correctly in the table1 and throughout the Ms.

Which ANOVA did you use for the stats?

sST2 is also produced by the lungs. What is the pulmonary status of those patients with exercise intolerance even if COPD have been excluded?

Results

Many had AF or LBBB; what is the impact on the results? sST2 is an independent predictor of death or HF in patients with AF irrespective of history of HF or NT-proBNP levels.

Table1: Surprisingly the more severe LVEF group had less oxidative which is counterintuitive for me. Would isoprostanes be a better index of OS? Moreover, later you found positive or negative correlations depending of the test which really questions the relevance of those tests.

Figure 1 (and all figures): show the individual datas, identify better the sST2 group (colored bar or like in Fig2), give the number of patient in each group

Table 2 : change , for .

Most of correlations are pretty poor. Correlations have been done on the whole population. Did you try the correlations by considering only the NYHa groups or LVEF groups? (I particular for sST2). Show as supplemental datas the significant correlations.

Minor

L64 : remove its

NT-proBNP or NT pro-BNP or NT proBNP   choose and correct

Comments on the Quality of English Language

There are cookies throughout the manuscript that require careful editing. Edition by a native English speaker is required.

Author Response

Dear Reviewer

Thank you for reading the work carefully and for your valuable comments.

The aim of this study was to analyse the relationship between the level of sST2, NT-proBNP and oxidative stress markers in patients with reduced ejection fraction (HFrEF) due to non-ischemic cardiomyopathy. A group of 88 patients with HFrEF were divided into four groups depending on left ventricle ejection fraction (≤ 25% and >25%) and NYHA class (I-II or III-IV) values. sST2 was not different between the groups while NT proBNP increased with the severity. sST2 had poor correlations with LA area and TAC that did not differ between the groups.

They conclude with “in stable patients with HF due to dilated cardiomyopathy, NT-proBNP, but not sST-2, was a useful predictor of LVEF based on biomechanical strain markers. Similarly, a high correlation was found between NT-proBNP and NYHA class. However, only sST-2 concentration correlated with oxidative stress, and the positive correlation with TAC is surprising. Further research and determination of larger amounts of enzymatic and non-enzymatic antioxidants as well as oxidation products seem necessary.”

Answer: Thank you for reading the work carefully and for your valuable comments. Of course, we plan to expand our research - increase the number of patients and determine more enzymatic and non-enzymatic antioxidants.

Many recent papers have studied sST2 and NT-proBNP in HFrEF. While NTproBNP is well established, sST2 is more recent and certainly looks for its relevance in the diagnostic area of HF in general. The use of a broad range population in the present did not help to find its possible position.

Abstract : Define TAC TOS and OSI

Answer: In abstract we have defined TAC, TOS and OSI.

Introduction

Give a definition of NYHA classification and the justification of clusterization of I-II and II-IV

Answer: We have done it

Define flozin and empagliflozin etc..

Answer: We have done it

LVEF is a poor echocardiographic index. The study would have been more interesting with the GLS. Do you have the GLS ?

Answer: Thank you for your comment. We know that left ventricular ejection fraction is a poor indicator of impairment of its function, but in this group of patients GLS was not measured due to the lack of software in our echocardiography.machine

Did you measure systemic arterial pressure ?

Answer: Yes, data was completed in the table 1 and 2 in manuscript.

LVEF ≥25%

N=49

LVEF <25%

N=39

P

Median

(Q1-Q3)

Median

(Q1-Q3)

Arterial pressure (systolic)

[mm Hg]

126,5

(112,0-142,8)

106,8

(93,00-126,0)

0,005

Arterial pressure (diastolic)

[mm Hg]

78,50

(69,50-88-50)

71,25

(64,50-80,00)

0,046

Arterial pressure (mean)

[mm Hg]

94,67

(84,75-105,6)

84.92

(71,67-93,33)

0,005

sST2

NT-proBNP

Spearman r

P

Spearman r

P

Arterial pressure (systolic)

[mm Hg]

-0,072

0,552

-0,496

P<0,001

Arterial pressure (diastolic)

[mm Hg]

0,165

0,171

-0,273

0,022

Arterial pressure (mean)

[mm Hg]

0,072

0,553

-0,377

P<0,001

Methods

Give the reference number of ethical committee and clinical trial

Answer: We have done it.

Define LBBB and RBBB

Answer: We have done it.

What is the echocardiographic machine? How many different sonographers?

Answer: We have done it.

Some medications, eg some beta-blockers, are known to have antioxidant properties? How did you take into account this point?

Answer: We took into account the treatment and  In our group The majority of patients were treated with β-blockers (96.6%) and either an angiotensin-converting enzyme (ACE) inhibitor (97.7%) or an angiotensin receptor blocker (36.4%) 

A control group without HF would have been of interest.

Answer: Thank you very much for your suggestion but we don’t have the control group

I supposed you measured sST2. Label correctly in the table1 and throughout the Ms.

Answer: We have done it.

Which ANOVA did you use for the stats?

Answer: ANOVA test was used for comparison the sST2 and NT-proBNP concentrations and redox parameters presented on Figures 1-3.

sST2 is also produced by the lungs. What is the pulmonary status of those patients with exercise intolerance even if COPD have been excluded?

Answer: The patients had no diagnosed pulmonary disease They were not treated for lung disease. They were assessed for qualifications for cardiac transplantation. The results of cardiopulmonary exercise tests confirmed dyspnea of cardiac origin

Results

Many had AF or LBBB; what is the impact on the results? sST2 is an independent predictor of death or HF in patients with AF irrespective of history of HF or NT-proBNP levels.

Answer: We have done it.   

LBBB (-)
AF (-)

LBBB (-)
AF (+)

LBBB (+) AF (-)

LBBB (+)
AF (+)

ANOVA
P

1

2

3

4

Median

(Q1-Q3)

Median

(Q1-Q3)

Median

(Q1-Q3)

Median

(minimum - maximum)

n

40

19

26

3

sST2,

ng/ml

15.41

(9,82-22,14)

16,12

(10,88-20,76)

19.06

(10,56-29,78)

30,94

(16.91-39,79)

0,272

NT-proBNP,

pg/ml

1093

(285,4-1682)

1505

(626-2351)

764

(281,0-2030)

2519

(122,0-4761)

0,439

 We added a sentence to the discussion:

In our study we did not demonstrate statistically significant differences in the concentration of sST2 (p=0.272) and NT-proBNP (p =0.439)in the subgroups of patients with AF and/or LBBB. It may be due fact that 21 of 29 patients with LBBB  received cardiac resynchronisation therapy.

Table1: Surprisingly the more severe LVEF group had less oxidative which is counterintuitive for me. Would isoprostanes be a better index of OS? Moreover, later you found positive or negative correlations depending of the test which really questions the relevance of those tests.

Answer: In subsequent study, we will consider the determination of isoprostanes as markers of oxidative stress. In our study group, we measured MDA. In the LVEF <25% group, the MDA concentration was 3.93±0.7 and was lower than in the LVEF >25% group (MDA = 4.65±1.6). However, the differences were not statistically significant.

Both study groups are groups of patients with dilated cardiomyopathy, i.e. with a disturbed oxidative-antioxidant balance. Lower TOS concentration and higher TAC concentration in the LVEF <25% group may be the result of greater stimulation of the enzymatic and non-enzymatic antioxidant system, resulting in lower TOS in this group. In the LVEF <25% group, we also observe higher concentrations of bilirubin and uric acid, which are non-enzymatic antioxidants.

Figure 1 (and all figures): show the individual datas, identify better the sST2 group (colored bar or like in Fig2), give the number of patient in each group

Answer: We have done it.

Table 2 : change , for .

Answer: We have done it.

Most of correlations are pretty poor. Correlations have been done on the whole population. Did you try the correlations by considering only the NYHa groups or LVEF groups? (I particular for sST2). Show as supplemental datas the significant correlations.

Answer: We added a sentence to the discussion:

In our study we demonstrated positively corralations between sST2 and TAC both in group EF <25% (r=0.367, p=0.021) and NYHA III-IV (r=0.534, p=0.002)  may suggest its relation to severity of the disease and low grade inflammation described HF.  

Supplemental dates

Correlations between examined clinical, echocardiographic, laboratory data and sST2 a NT-proBNP concentration.

sST2

LVEF<25%

sST2

LVEF>25%

NT-proBNP

LVEF<25%

NT-proBNP

LVEF>25%

Spearman r

P

Spearman r

P

Spearman r

p

Spearman r

p

Age, years

-0,031

0,850

0,203

0,161

-0,280

0,084

-0,275

0,056

BMI, kg/m2

-0,411

0,009

0,330

0,021

-0,376

0,018

-0,321

0,025

NYHA, I-IV

0,073

0,657

0,145

0,319

0,553

0,000

0,425

0,002

LVEF, %

-0,080

0,629

0,120

0,410

-0,100

0,545

-0,261

0,071

LVEDV, ml

-0,185

0,260

-0,108

0,461

-0,248

0,128

0,475

0,001

LVEDD, mm

-0,303

0,060

-0,145

0,320

-0,133

0,419

0,348

0,014

LA area, cm2

0,174

0,296

-0,125

0,402

0,282

0,086

0,500

0,000

RA area, cm2

-0,008

0,960

-0,017

0,908

0,402

0,012

0,379

0,009

NT-proBNP, pg/ml

0,435

0,006

-0,146

0,316

sST2, ng/ml

0,435

0,006

-0,146

0,316

CRP, mg/l

0,035

0,861

0,194

0,304

0,001

0,995

-0,387

0,034

Interleukin -6 pg/ml

0,460

0,008

-0,240

0,147

0,286

0,113

-0,089

0,594

eGFR, ml/min

-0,267

0,100

0,038

0,797

-0,351

0,028

0,087

0,554

sodium, mmol/l

-0,265

0,103

0,297

0,038

-0,294

0,069

0,014

0,921

hemoglobin, g/dl

-0,259

0,112

0,380

0,007

-0,206

0,209

-0,185

0,203

AspAT,  IU/l

-0,356

0,026

0,145

0,319

0,080

0,627

-0,072

0,625

AlAT, IU/l

-0,340

0,034

0,220

0,128

-0,046

0,780

-0,162

0,265

uric acid, μmol/l

0,265

0,103

0,151

0,301

0,347

0,030

0,186

0,200

bilirubin, µmol/l

0,140

0,394

0,206

0,160

0,316

0,050

0,279

0,055

TOS, µmol/l

-0,273

0,093

0,163

0,263

-0,210

0,200

0,086

0,557

TAC, mmol/

0,367

0,021

0,117

0,424

0,090

0,586

-0,101

0,488

OSI

-0,304

0,060

0,023

0,875

-0,201

0,220

0,193

0,185

BMI, Body Mass Index; NYHA, New York Heart Association functional class; LVEF, left ventricle ejection fraction; LVEDV, ventricular end-diastolic volume; LVEDD, left ventricular end-diastolic diameter; LA area, left atrial area; RA area, right atrial area; NT-proBNP, N-terminal pro-B-type natriuretic peptide; sST2, Soluble suppression of tumorigenicity 2 protein; CRP, C-reactive protein; eGFR, estimated glomerular filtration rate; AspAT, aspartate aminotransferase; AlAT, alanine aminotransferase; TOS, total oxidant status; TAC, total antioxidant capacity; TOS,

Correlations between examined clinical, echocardiographic, laboratory data and sST2 a NT-proBNP concentration.

sST2

NYHA III-IV

sST2

NYHA I-II

NT-proBNP

NYHA III-IV

NT-proBNP

NYHA I-II

Spearman r

P

Spearman r

P

Spearman r

P

Spearman r

P

Age, years

-0,023

0,902

0,175

0,192

-0,355

0,050

-0,269

0,043

BMI, kg/m2

-0,134

0,473

0,027

0,839

-0,345

0,058

-0,391

0,003

NYHA, I-IV

0,112

0,548

0,145

0,282

0,405

0,024

0,151

0,263

LVEF, %

-0,066

0,726

-0,022

0,868

-0,137

0,464

-0,444

0,001

LVEDV, ml

-0,068

0,716

-0,139

0,302

-0,007

0,971

0,307

0,020

LVEDD, mm

-0,299

0,102

-0,167

0,214

-0,064

0,734

0,259

0,051

LA area, cm2

0,091

0,638

-0,020

0,885

0,272

0,153

0,427

0,001

RA area, cm2

0,084

0,665

-0,065

0,635

0,241

0,207

0,435

0,001

NT-proBNP, pg/ml

0,225

0,224

0,019

0,886

sST2, ng/ml

0,225

0,224

0,019

0,886

CRP, mg/l

0,173

0,419

0,127

0,480

-0,169

0,431

-0,345

0,049

Interleukin -6 pg/ml

0,574

0,002

-0,280

0,069

0,119

0,554

0,128

0,412

eGFR, ml/min

-0,145

0,436

-0,191

0,155

-0,300

0,102

-0,191

0,155

sodium, mmol/l

-0,024

0,896

0,053

0,694

-0,292

0,110

-0,057

0,672

hemoglobin, g/dl

0,016

0,930

0,154

0,253

-0,196

0,291

-0,109

0,421

AspAT,  IU/l

-0,183

0,325

0,013

0,921

-0,061

0,744

0,011

0,938

AlAT, IU/l

-0,282

0,124

0,109

0,421

-0,101

0,588

-0,074

0,583

uric acid, μmol/l

0,354

0,051

0,144

0,286

0,427

0,017

0,264

0,047

bilirubin, µmol/l

-0,028

0,882

0,276

0,038

0,017

0,928

0,212

0,113

TOS, µmol/l

-0,222

0,230

-0,075

0,578

-0,186

0,316

-0,227

0,089

TAC, mmol/

0,534

0,002

0,101

0,453

0,244

0,186

-0,029

0,831

OSI

-0,279

0,128

-0,155

0,192

-0,222

0,231

-0,061

0,650

BMI, Body Mass Index; NYHA, New York Heart Association functional class; LVEF, left ventricle ejection fraction; LVEDV, ventricular end-diastolic volume; LVEDD, left ventricular end-diastolic diameter; LA area, left atrial area; RA area, right atrial area; NT-proBNP, N-terminal pro-B-type natriuretic peptide; sST2, Soluble suppression of tumorigenicity 2 protein; CRP, C-reactive protein; eGFR, estimated glomerular filtration rate; AspAT, aspartate aminotransferase; AlAT, alanine aminotransferase; TOS, total oxidant status; TAC, total antioxidant capacity; TOS,

Minor

L64 : remove its

Answer: We have done it.

NT-proBNP or NT pro-BNP or NT proBNP   choose and correct

 Answer: We have done it.

Comments on the Quality of English Language

There are cookies throughout the manuscript that require careful editing. Edition by a native English speaker is required.

Answer: We have done English correction.

Round 2

Reviewer 2 Report

Comments and Suggestions for Authors

Changes made by the authors in the manuscript have not been identified. This is difficult to review.

The cause of DCM is not detailed thus there are many confounding factors thus for all figures insert the individual data to evaluate heterogeneity. Table 1 remove at least one digit for the echo data

Arterial pressure: give the methodology (what is the heart rate at the moment of the measurement?).

Echocardiography: Who performed the echo? How many different sonographers and who perfomed the analysis? Was it always the same person?

Precise for each figure/Table the test used (oneway/two-way). Statistics should be verified. I don’t believe you can use a simple KW ANOVA in Fig 1-3.

Author Response

Changes made by the authors in the manuscript have not been identified. This is difficult to review.

Answer: We have marked all changes made in red.

The cause of DCM is not detailed thus there are many confounding factors thus for all figures insert the individual data to evaluate heterogeneity. Table 1 remove at least one digit for the echo data

Answer: We removed digit for the echo data.

In the study group there were two cases of familial cardiomyopathy.The rest was cardiomyopathy of idiopathic etiology

Arterial pressure: give the methodology (what is the heart rate at the moment of the measurement?).

Answer:

Heart rate:

Median - 73,4; Q-64,8; Q2-82,8; min/max 48-118/min; mean- 74,5±13,62

LVEF >25  median 66,5; Q1-62,4;  Q2-81,2

LVEF<25  median - 77,0; Q1-68,0; Q2-84,8

p< 0,008

We added this information to the table 1.

Arterial pressure was measured according to the Korotkoff method. We give this information in methodology section.

Echocardiography: Who performed the echo? How many different sonographers and who perfomed the analysis? Was it always the same person?

Answer:

Echocardiograms were assessed by two experienced investigators independently (interobserver variability was <8%). The final decision was made through consensus. It was the same two person.

Which ANOVA did you use for the stats?

Precise for each figure/Table the test used (oneway/two-way). Statistics should be verified. I don’t believe you can use a simple KW ANOVA in Fig 1-3.

Answer:

Kruskal–Wallis one-way analysis of variance (one-way ANOVA on ranks) was used.

The “Statistics” chapter and descriptions of figures attached: